# Dry Friction Performances of MoN$_x$ Coatings Deposited by High–Power Pulsed Magnetron Sputtering

**Fuqiang Li, Wei Dai \*** , **Qimin Wang \***, **Haiqing Li and Zhengtao Wu**

School of Electromechanical Engineering, Guangdong University of Technology, Guangzhou 510006, China
\* Correspondence: weidai@gdut.edu.cn (W.D.); qmwang@gdut.edu.cn (Q.W.)

**Abstract:** A MoN$_x$ coating serves as an effective wear protection layer and is crucial for the investigation of its tribological characteristics at various temperatures. This study examined the tribological characteristics of MoN$_x$ coatings that were deposited through high-power pulsed magnetron sputtering (HiPIMS) in an Ar/N$_2$ environment with varying N$_2$ partial pressures. The microstructures and mechanical properties of the coatings were elucidated using scanning electron microscopy, grazing-incidence-angle X-ray diffraction, energy-dispersive spectroscopy, and nanoindentation. The dry friction performances of the coatings at different heating temperatures were studied using a ball-on-disk tribometer. The MoN$_x$ coating produced by HiPIMS was composed primarily of fcc$-$Mo$_2$N and featured a fine, dense column crystal with a maximum hardness of 28.8 GPa. The MoN$_x$ coatings exhibited excellent lubrication and wear reduction properties at room temperature (RT). The dry friction performances of the MoN$_x$ coatings at elevated temperatures were expected to depend on the growth of the MoO$_3$ tribolayer. At relatively low temperatures (300 °C and 400 °C), the MoO$_3$ tribolayer grew slowly and was not enough to provide good lubrication, causing increases in the dry friction of the coatings. However, the δ$-$MoN phase formed in the MoN$_x$ coating deposited at a high N$_2$ partial pressure could facilitate the formation of MoO$_3$ and thus decreased the friction coefficient at 400 °C. At the relatively high heating temperature of 500 °C, however, the MoO$_3$ tribolayer grew so rapidly that the oxide layer became thick, resulting in an increase in the wear rate. It is believed that tuning the growth rate of MoO$_3$ via optimizing the composition and structure of the MoN$_x$ coatings might be a useful way to improve the dry friction at various elevated temperatures.

**Keywords:** molybdenum nitride; mechanical properties; dry friction; wear

## 1. Introduction

Owing to their high hardness and wear resistance, metal nitride coatings (e.g., TiN and CrN, which are commonly available) can be used as protective coatings for mechanical parts to reduce surface damage to equipment and improve its service life [1,2]. However, in addition to wear protection, mechanical components also require a low coefficient of friction to improve transmission efficiency; moreover, frictional wear is one of the main causes of energy loss in automobiles [3]. Therefore, applying metal nitride wear-resistant coatings to the surfaces of friction subsets can help achieve energy savings and promote green development. MoN$_x$ coatings reduce the coefficient of friction by forming easily sheared layers of molybdenum oxide on the friction contact surface [1,4]. Although the wear and friction reduction effect of MoN$_x$ coatings is slightly worse than those of DLC and diamond coatings, MoN$_x$ coatings can adapt to shock vibrations and high-temperature working environments. DLC coatings have poor thermal temperature, poor bonding, and brittleness. The preparation of diamond coatings requires high-temperature conditions and incurs high costs. In comparison, a metal nitride coating may not only reduce friction and wear, but it is also easier to create, less expensive, and more reliable.

The excellent mechanical and frictional properties of MoN$_x$ coatings make them suitable for wear-resistant lubrication and protective coating. However, MoN$_x$ coatings have

poor tribological performance at higher temperatures, mainly due to the loose formation of oxides and their volatilization at high temperatures (350–400 °C) [5]. The common preparation methods for $MoN_x$ coatings are arc ion plating [6,7], reactive magnetron sputtering [8,9], high-power impulse magnetron sputtering (HiPIMS) [9], and ion-beam-assisted deposition [10]. $MoN_x$ coatings exhibit different phase structures and properties, depending on their deposition conditions. Tetragonal $\beta-Mo_2N$, cubic $\gamma-Mo_2N$, and hexagonal $\delta-MoN$ are the most reported phase structures of $MoN_x$ coatings [6,11–17]. Other phase structures such as orthorhombic $\varepsilon-Mo_4N_3$, cubic MoN, monoclinic $\sigma-MoN$, and hexagonal $\delta-MoN$ have also been detected [11].The different phase structures of $MoN_x$ coatings can be obtained by changing the process parameters; for example, the two-phase combination of $\delta-MoN$ and $\gamma-Mo_2N$ exhibits better tribological properties [6,18]. Furthermore, the pure $\gamma-Mo_2N$ structure of $MoN_x$ coatings exhibits a hardness of 22.2 GPa, which increases to 25.2 GPa because of the simultaneous presence of the $\gamma-Mo_2N$ and $\delta-MoN$ phases [19]. Different crystal structures are obtained by varying the $N_2$ partial pressure during the magnetron sputtering process [17]. Moreover, varying the substrate temperature and the species energy during deposition affects the crystal structure of vapor-deposited coatings [6,20,21]. While nanocomposite coatings with lower friction coefficients, such as MoCuN and MoAgN, can also be obtained via elemental doping, MoSiN coatings are considered to have better wear resistance [22–24]. We may further enhance $MoN_x$'s tribological properties, both in terms of elemental doping and structural design, by studying the tribological characteristics of distinct phases of the material at various temperatures. The HiPIMS process can reduce coating roughness, improve the $N_2$ dissociation reaction, and refine the coating structure, thereby improving the wear and friction reduction effect of $MoN_x$ coatings compared with those prepared through conventional magnetron sputtering. The mechanical properties of $MoN_x$ coatings prepared through the HiPIMS process under different $N_2$ partial pressures and the frictional properties of these coatings at different temperatures have not yet been systematically investigated [25–27].

For the further development and application of $MoN_x$ coatings, the dry tribological properties of $MoN_x$ coatings prepared at different temperatures using the HiPIMS technology and the effects of different $N_2$ partial pressures on their tribological properties should be systematically investigated. In this study, we utilized the high plasma density provided by HiPIMS to prepare $MoN_x$ coatings at different $N_2$ partial pressures. A denser grain structure was obtained through the HiPIMS process, and the resulting surface did not contain any droplet particles due to arc ion plating. Furthermore, the coating exhibited high adhesion and low surface roughness, thereby improving the coating quality [28]. The effects of different $N_2$ partial pressures on the mechanical properties and structures of $MoN_x$ coatings were investigated. In addition, the tribological properties of $MoN_x$ coatings prepared at different $N_2$ partial pressures were systematically analyzed at room temperature (RT), 300 °C, 400 °C, and 500 °C.

## 2. Experimental Details

### 2.1. Coating Deposition

The $MoN_x$ coatings were deposited on AISI304 stainless-steel sheets (50 mm × 10 mm × 0.8 mm) and cemented carbide blocks (WC-15 wt.% TiC-6 wt.% Co, 16 mm × 16 mm × 4 mm) using HiPIMS with a pure Mo (99.95%) target. The substrates were ultrasonically cleaned with anhydrous ethanol for 30 min before deposition. The deposition chamber was heated to 350 °C and evacuated to $7 \times 10^{-3}$ Pa. Subsequently, Ar (99.999% purity) was introduced under a pressure of 2.0 Pa, and the glow discharge etching process was conducted at a bias voltage of −1000 V for 30 min to remove surface contaminants. During 2 h of deposition, the HiPIMS power supply was fixed at 2.0 kW with a pulse frequency of 300 Hz and a pulse width of 100 μs. In addition, the bias voltage was held constant at −100 V, and the total pressure of $N_2$ and Ar was 0.6 Pa. $N_2$ partial pressures of 0.12, 0.24, 0.36, and 0.48 Pa were applied to obtain various coating samples. For simplicity, the

samples were labeled as $MoN_x$_1#, $MoN_x$_2#, $MoN_x$_3#, and $MoN_x$_4#, corresponding to the $N_2$ partial pressures of 0.12, 0.24, 0.36, and 0.48 Pa, respectively.

## 2.2. Coating Characterization

The coating cross-sectional morphologies and elemental compositions were characterized using scanning electron microscopy (SEM, Hitachi SU8220) with an operating voltage of 10 kV and energy-dispersive X-ray spectroscopy (EDS, Oxford Instruments X-Max$^N$). After the friction tests, grazing-incidence X-ray diffraction (GIXRD, Bruker D8 Advance) was performed to determine the phase structures of the $MoN_x$ coatings in the as-deposited state using a Cu Kα radiation source operated at 40 kV and 40 mA. The X-ray diffractograms were recorded at an incident angle of 1.0°, a scanning step of 0.02°, and a dwell time of 1.0 s. The substrate curvature method based on Stoney's equation (Equation (1)) was used to experimentally measure the residual stresses of the coatings with a film stress tester (FST-1000, Supro Instruments, Shenzhen, China) [29].

$$\sigma_s = \frac{E_s}{6(1 - v_s)} \frac{h_s^2}{h_c} \left( \frac{1}{R} - \frac{1}{R_0} \right) \tag{1}$$

where $E_s$ and $v_s$ refer to the elastic modulus and Poisson's ratio of the AISI304 stainless−steel sheet (195.6 GPa and 0.29), and $h_s$ and $h_c$ denote the thicknesses of the substrates and coatings, respectively. $R_0$ and $R$ denote the curvature radii of the substrates before and after deposition, respectively.

The hardness and elastic modulus values of the $MoN_x$ coatings were measured using a nanoindentation tester (Anton Paar TTX-NHT$^2$) and a Berkovich diamond indenter via the Oliver and Pharr method [30]. The indentation depth of the indenter was kept below 10% of the coating thickness to reduce the influence of the substrate on the test results. The force was loaded and unloaded at a rate of 20 mN/min, with a holding time of 10 s and a peak load of 10 mN. Tribological tests were performed at RT (relative humidity <50%), 300 °C, 400 °C, and 500 °C using a ball-on-disc tribometer (Anton Paar HT1000, Graz, Austria), with $Al_2O_3$ balls (Ø6.0 mm) as the counterpart. A normal load of 5 N was applied to the friction tests at RT, whereas a load of 2 N was selected for tests at higher temperatures. The radii of the wear tracks were selected as 2 mm for all friction tests. The variations in the friction coefficient with the sliding lap were recorded at a constant sliding speed of 0.1 m/s. The wear tracks were further analyzed using SEM and confocal laser scanning microscopy (Olympus OLS4100, Tokyo, Japan). The wear rates of the coatings were calculated according to the Archard equation (Equation: k = V/(F × L)), where k denotes the wear rate ($mm^3$/N·m), V denotes the wear volume ($mm^3$), F denotes the normal load (N), and L denotes the total sliding distance (m). Detailed microstructural investigations of the deposited state of the $MoN_x$_4# coating and the tribolayer after friction at 400 °C were carried out using transmission electron microscopy (TEM) and scanning transmission electron microscopy (STEM, Thermo Fisher Talos F200S) with a field emission gun operating at 200 kV. The TEM sample was prepared using focused ion beam scanning electron microscopy (Thermo Fisher Scios2, Waltham, MA, USA) following the lift−out procedure. A final surface cleaning was conducted at 5 kV and 16 pA to minimize the damage and artifacts induced by the ion beam milling. Raman spectrometers (LabRAM HR Evolution 532 nm lasers) were also used to determine the compositions of the tribolayers and the oxidation of the coatings.

## 3. Results and Discussion

### 3.1. Microstructure and Mechanical Properties

The chemical compositions of the $MoN_x$ coatings deposited under different $N_2$ partial pressures are listed in Table 1. With the gradual increase in the $N_2$ partial pressure, more nitrogen was involved in the reaction and the N content of the coating increased. Figure 1 displays the fracture cross-sections of the $MoN_x$ coatings. The $MoN_x$_1# coating, with a N content of 27.5 at.%, exhibited dense and fine grain growth. In contrast, a typical columnar

grain morphology was exhibited by the $MoN_x$_2#, _3#, and _4# coatings. Furthermore, the increase in $N_2$ partial pressure caused significant target poisoning and decreased the deposition rate. The $MoN_x$_1#, _2#, _3#, and _4# coatings had thicknesses of 2.3, 2.4, 1.9, and 1.4 µm, respectively.

**Table 1.** Elemental compositions and thickness, residual stress, hardness, and elastic modulus values of $MoN_x$ coatings deposited through HiPIMS under different $N_2$ partial pressures.

| Samples | $N_2$ Partial Pressure (Pa) | Elemental Content (at.%) | | Thickness (µm) | Residual Stress (GPa) | Hardness (GPa) | Elastic Modulus (GPa) |
|---|---|---|---|---|---|---|---|
| | | **Mo** | **N** | | | | |
| 1# | 0.12 | 72.5 | 27.5 | 2.3 ± 0.2 | −2.7 ± 0.1 | 24.7 ± 0.7 | 394 ± 6 |
| 2# | 0.24 | 67.5 | 32.5 | 2.4 ± 0.3 | −4.3 ± 0.2 | 26.2 ± 0.6 | 439 ± 8 |
| 3# | 0.36 | 60.3 | 39.7 | 1.9 ± 0.2 | −4.9 ± 0.2 | 28.8 ± 0.6 | 482 ± 7 |
| 4# | 0.48 | 54.1 | 45.9 | 1.4 ± 0.1 | −3.3 ± 0.2 | 26.2 ± 0.6 | 452 ± 9 |

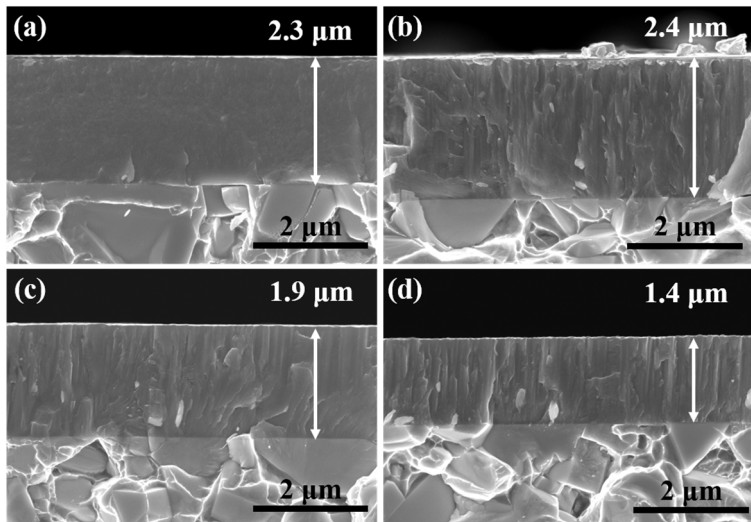

**Figure 1.** SEM cross−sectional morphologies of $MoN_x$ coatings deposited under different $N_2$ partial pressures: (**a**) $MoN_x$_1#, (**b**) $MoN_x$_2#, (**c**) $MoN_x$_3#, and (**d**) $MoN_x$_4#.

Figure 2 illustrates the phase determination for the $MoN_x$ coatings through GIXRD; the $MoN_x$ coatings contained a body−centered cubic Mo metallic phase (space group $Im\bar{3}m$, ICDD 00-042-1120). Broad diffraction peaks were detected for the $MoN_x$_1# coatings, indicating a small grain size, which was consistent with the aforementioned SEM observations. A nitride phase was observed in $MoN_x$_1#, which exhibited poor crystallization and an even, amorphous structure, thereby resulting in poor detection by XRD. The competitive growth of metallic and nitride phases resulted in very fine grains of the $MoN_x$_1# coating. With the increase in the $N_2$ partial pressure, a cubic $Mo_2N$ structure (space group $Pm\bar{3}m$, ICDD 00-025-1366) was obtained for the $MoN_x$_2#, _3#, and _4# coatings. Furthermore, the increase in the N content resulted in an increased lattice parameter in the $Mo_2N$ phase, leading to a shift in the diffraction peaks to small Bragg angles. According to a previous study [31], the Mo−N system contained two bcc phases: $\gamma−Mo_2N$ (with lattice parameters ranging from 4.16 to 4.19 Å) and B1−MoN (with lattice parameters ranging from 4.20 to 4.27 Å). The XRD peaks of the prepared $MoN_x$ coatings were notably shifted to lower angles with the increase in the $N_2$ partial pressure, likely caused by the solid solution of N or phase transformation; both indicate increases in the lattice parameters of the coating. Because the peak positions of these $MoN_x$ phases are considerably close to each other, distinguishing between them is challenging. According to the experimental results and those reported in other studies, an increase in the $N_2$ partial pressure primarily leads to the following phase transition processes: bcc $\alpha−Mo \rightarrow$ fcc $\gamma−Mo_2N/\beta−Mo_2N \rightarrow B1−MoN$ [32].

Furthermore, some researchers prepared coatings with δ−MoN as the main phase at a $N_2$ partial pressure of 0.43 Pa [33]. Similar to N-saturated fcc MoN, hexagonal δ−MoN may form with a continued increase in the $N_2$ partial pressure. The Scherrer equation was used to determine the average grain size of the $MoN_x$ coating, and the results showed that the coating's grain size varied as a function of the $N_2$ partial pressure as follows: 3 nm (0.12 Pa), 9.9 nm (0.24 Pa), 14.8 nm (0.36 Pa), and 13.4 nm (0.48 Pa). $MoN_x\_1\#$ was due to the metallic phase α−Mo and fcc−$Mo_2N$ competitive growth resulting in crystallization, while HIPIMS high-energy particle bombardment was also conducive to crystallization. As the nitrogen division increased, more nitrogen participated in the reaction, the metal phase α−Mo disappeared, and the $MoN_x$ crystal grew. Additionally, the grain size of $MoN_x\_4\#$ matched that of the high-resolution TEM image (Figure 3c).

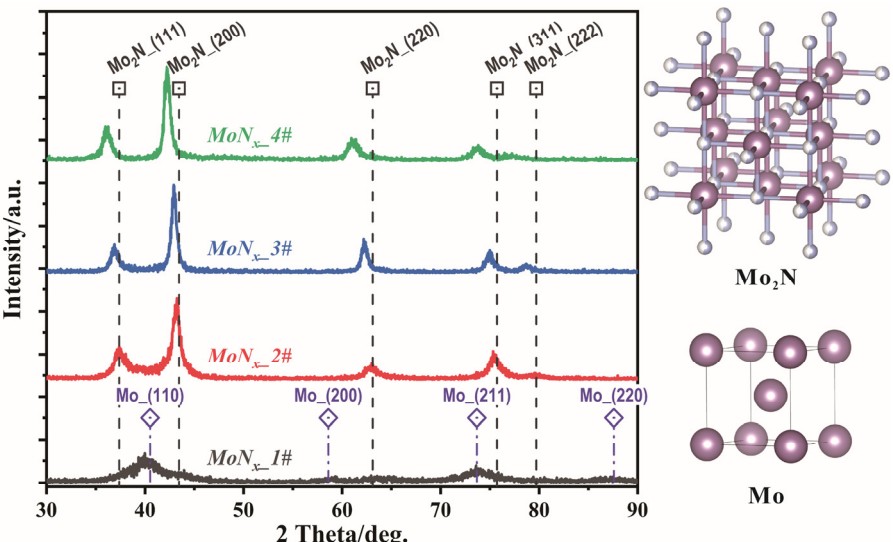

**Figure 2.** GIXRD patterns of $MoN_x$ coatings deposited through HiPIMS under different $N_2$ partial pressures.

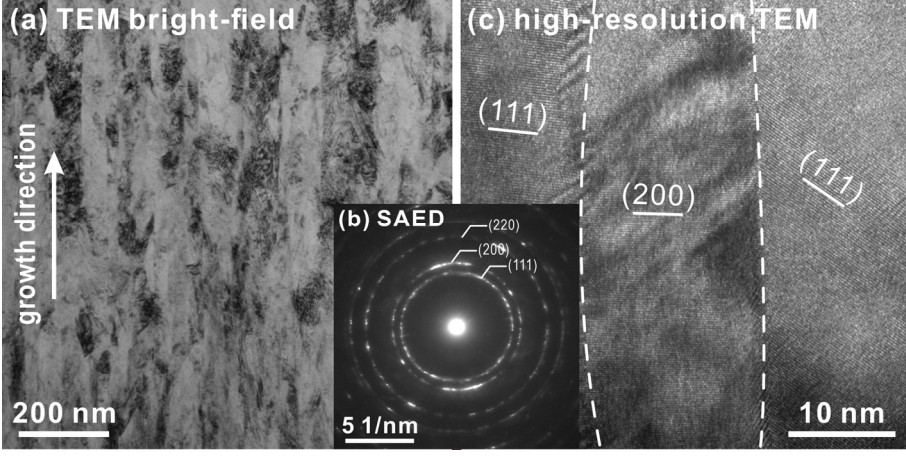

**Figure 3.** (**a**) Bright−field TEM image, (**b**) selective−area electron diffraction pattern, and (**c**) high-resolution TEM image of the $MoN_x\_4\#$ coating.

Figure 3 illustrates the TEM results of the $MoN_x\_4\#$ coating. A columnar grain growth with a width of dozens of nanometers can be viewed more clearly in the bright-field TEM image (Figure 3a). The selective-area electron diffraction pattern (Figure 3b) indicates a cubic $Mo_2N$ structure, which is consistent with the XRD results. Figure 3b shows distinct (111), (200), and (220) diffusion rings, which are equivalent to the fcc−$Mo_2N$ diffraction rings in GIXRD. Adjacent columnar grains with varied orientation are indicated the lattice-

resolved high-resolution micrograph (Figure 3c). In Figure 3c, the lattice stripes of (111) and (200) are well aligned, and the grain boundaries have partial co-lattice interfaces.

The residual stresses of the $MoN_x$ coatings with a compressive stress state were calculated using the Stoney equation, as displayed in Table 1. With the increase in the $N_2$ partial pressure from 0.12 to 0.36 Pa, the residual stress gradually increased from −2.7 to −4.9 GPa. The increase in the residual stress occurred mainly because of the phase change of the as-deposited coatings from the metallic to the nitride phase, with the formation of well-defined columnar grains. However, when the $N_2$ partial pressure was increased to 0.48 Pa, the residual stress of $MoN_x$_4# decreased to −3.3 GPa. This reduction was related to the reduced ion bombardment during deposition under higher $N_2$ partial pressures [34] and the decrease in the coating thickness. With the increase in the coating thickness, growth stress occurred as a part of the total residual stress.

Owing to the dense structure and the presence of fine grains, $MoN_x$_1# exhibited a hardness of 24.7 GPa, which was significantly higher than that of pure metallic coatings [19]. Compared with conventional magnetron sputtering, HiPIMS achieves better mechanical properties under low $N_2$ partial pressures, probably due to the effect of crystallization. The transformation from a metal-dominated phase structure to nitride caused an increase in hardness from the 24.7 GPa of $MoN_x$_1# to the 28.8 GPa of $MoN_x$_3#. The reduced hardness (26.2 GPa) of the $MoN_x$_4# coating can be attributed to decreased compressive residual stress. The elastic modulus changed in proportion with the hardness; it first increased to 482 GPa ($MoN_x$_3#) and then slightly decreased to 452 GPa ($MoN_x$_4#). Moreover, the decreasing trend of the hardness of the as−deposited $MoN_x$ coatings was related to the intrinsic hardening mechanism of the increased lattice parameters [8]. Under low $N_2$ partial pressures, the 4d sublevel electrons in Mo and 2p sublevel electrons in N form Mo–N covalent bonds from the lower energy band. With the increase in the $N_2$ partial pressure, the electrons in the lower bond band are accompanied by additional electrons; this results in the breakage of the covalent bonds and the formation of weak ionic bonds, thereby causing a decrease in hardness [35]. Covalent bonding may lead to a high bulk modulus and mechanical hardness, and $MoN_x$ coatings with higher residual compressive stresses exhibit higher hardness values. In addition, residual compressive stress relaxation can reduce hardness [36]. It is also possible that at a $N_2$ partial pressure of 0.48 Pa there is a small amount of δ−MoN in the coating, so the mechanical properties and residual stresses are reduced. The XRD peaks of the prepared $MoN_x$ coatings were notably shifted to lower angles.

### *3.2. Tribological Properties*

The tribological properties of the $MoN_x$ coatings were measured at RT and higher temperatures (T = 300, 400, and 500 °C). Figure 4 displays the coefficients of friction (COFs) of the $MoN_x$ coatings after the friction tests at (a) RT, (b) 300 °C, (c) 400 °C, and (d) 500 °C. The average friction coefficients and wear rates of the $MoN_x$ coatings are displayed in Figure 5. The $MoN_x$ coatings exhibited advantages such as a low COF and a low wear rate at RT; the low COF of $MoN_x$ coatings at RT (~0.32) occurred because of the formation of a $MoO_3/H_2O$ tribolayer that exhibited superior lubrication properties, similar to $WN_x$ coatings [37,38]. In particular, the wear rate of $MoN_x$_4# was only $9.4 \times 10^{-9}$ mm$^3$/N·m. The wear trace curve displayed in Figure 6d indicates a large accumulation of abrasive chips inside the wear trace, thereby effectively reducing the wear of the coating. At 300 °C, the COF of the coatings was higher, around 0.6. It is clear that the increase in the COF is attributed to the evaporation of water. The tribological behavior of the coating changed to dry friction. The tribolayer did not have the same friction-reducing lubricating effect as at RT. However, all the coatings showed good wear resistance at 300 °C, similar to that at RT. At 400 °C, the COF of the coatings began to differ and increased compared to that at RT; the COF of $MoN_x$_1# increased to 0.68, that of $MoN_x$_4# increased to 0.39, and those of $MoN_x$_2# and $MoN_x$_3# were relatively similar. $MoN_x$_1# contained a large amount of metallic Mo phase and exhibited a slightly larger wear rate than $MoN_x$_4#, which exhibited

the lowest wear rate of $3 \times 10^{-7}$ mm$^3$/N·m. At 500 °C, the coatings had low COFs, with those of MoN$_x$_1# and 2# close to 0.4, likely because of the liquefaction of the generated oxides. However, all the coatings underwent severe wear, and the wear rate of the least worn MoN$_x$_3# was as high as $1.1 \times 10^{-5}$ mm$^3$/N·m. The COFs of the various MoN$_x$ coatings became progressively smaller at 500 °C with variations in the friction process, and the differences in the wear rates were small. Some researchers have used conventional magnetron sputtering to prepare MoN$_x$ coatings at different nitrogen flow rates [8], and the lowest COF has been reported to be 0.5 at RT. The COF of the MoN$_x$ coating that contained the metallic Mo phase was as high as 0.8, which was notably inconsistent with the tribological properties of the MoN$_x$ coatings prepared via HiPIMS at RT in this study. The tribological properties of the coatings were considerably improved by HiPIMS.

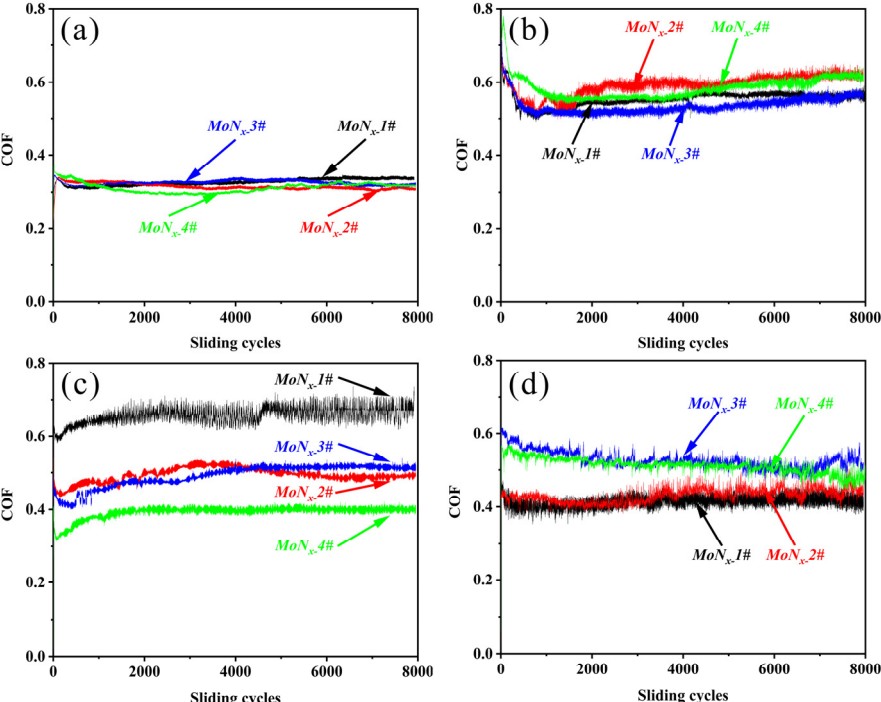

**Figure 4.** Friction coefficients of MoN$_x$ coatings after friction tests at (**a**) RT, (**b**) 300 °C, (**c**) 400 °C, and (**d**) 500 °C.

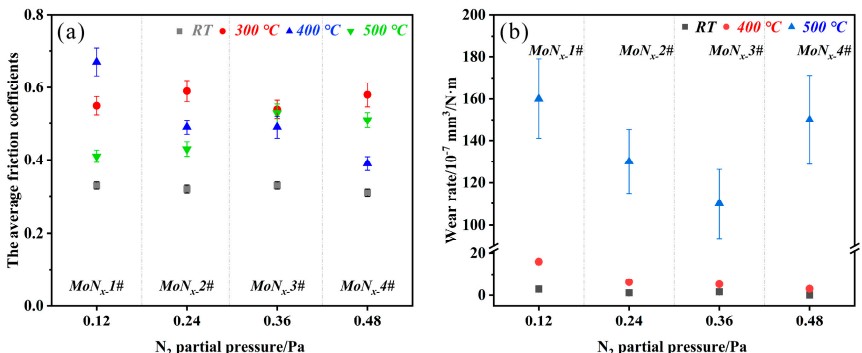

**Figure 5.** Average friction coefficients (**a**) and wear rates (**b**) of MoN$_x$ coatings.

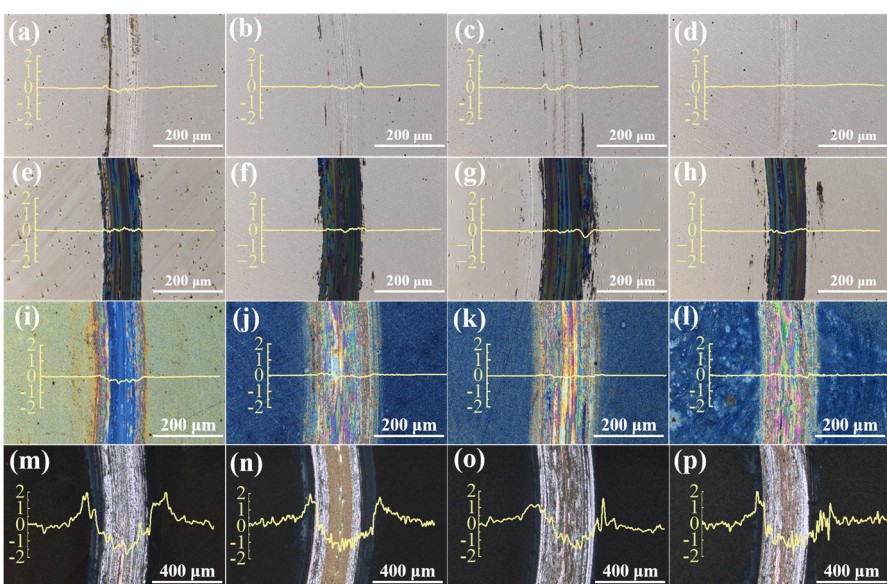

**Figure 6.** Surface morphologies and wear track cross-sectional profiles of the (**a**,**e**,**i**,**m**) MoN$_x$_1#, (**b**,**f**,**j**,**n**) _2#, (**c**,**g**,**k**,**o**) _3#, and (**d**,**h**,**l**,**p**) _4# coatings after friction tests at (**a**−**d**) RT, (**e**−**h**) 300 °C, (**i**−**l**) 400 °C, and (**m**−**p**) 500 °C.

Figure 6 displays the surface morphologies and wear track cross-sectional profiles of the (a, e, i, and m) MoN$_x$_1#, (b, f, j, and n) _2#, (c, g, k, and o) _3#, and (d, h, l, and p) _4# coatings after friction tests at (a−d) RT, (e−h) 300 °C, (i−l) 400 °C, and (m−p) 500 °C. The surfaces of the samples were the same as room temperature after the friction test at 300 °C, and insignificant oxidation occurred. The oxidation inside the abrasion marks was more obvious than at room temperature. More serious abrasive wear of the coating occurred in both temperature conditions, and the ploughing inside the abrasion marks was obvious. After the friction tests at 400 and 500 °C, a difference in the color of the coating surface was evident; the surface appeared blue after the friction test at 400 °C and black after the friction test at 500 °C. At RT, the wear marks of MoN$_x$_4# were the narrowest, and abrasive chips accumulated inside the wear marks, as displayed in the cross-section in Figure 6d. The wear of the MoN$_x$_4# coating was minimal, whereas the wear of the MoN$_x$_1# coating was slightly larger, and the wear marks were deeper, likely because of the increased Mo phase in the coating. At RT, the coating mainly underwent abrasive wear, with an accumulation of abrasive chips at the edges of the wear marks. According to Soignard et al., the compressibility of the MoN$_x$ phase is lower than that of pure metal [39]. Thus, MoN$_x$_1# was more easily deformed on the friction contact surface, and the metal was easily bonded and more susceptible to wear. The accumulation of abrasive chips in the wear marks behaved as a lubricant and reduced wear, making MoN$_x$_4# almost wear-free.

After the friction test at 400 °C, colored Magnéli (oxide) phases were formed inside the wear marks of all samples. MoN$_x$_1# exhibited darker wear marks and less oxidation on the sample surface (the other samples had blue surfaces). We observed a large number of white spots on MoN$_x$_4# that may have been traces of the volatilization of MoO$_3$. At 400 °C, the oxidation of the MoN$_x$ coatings was relatively low, but MoN$_x$_4# was more prone to oxidation, especially within the wear marks, and thus had lower COF and wear rate values. The XRD peaks for the MoN$_x$ coatings prepared under high N$_2$ partial pressures shifted to a lower angle and overlapped with the δ−MoN peak position, whereas MoN$_x$_4# exhibited more oxidation susceptibility in friction experiments at 400 °C, likely because δ−MoN was more easily oxidized to form a lubricant phase [40,41]. The susceptibility of the MoN$_x$ coatings to oxidation at RT and at 400 °C induced notable effects on their tribological properties. At 500 °C, the surfaces of the samples were dark gray, and large accumulations of abrasive chips occurred at the edges of the abrasion marks. The wear and abrasion marks in the MoN$_x$ coatings exhibited sharp increases. The generated MoO$_3$ liquefied

and volatilized at 500 °C [42], and the oxide layer formed on the surface of the specimen was exceedingly thick and fragile. It easily deformed upon contact with the grinding ball and accumulated at the edges of the abrasion marks after being squeezed. Under these conditions, the coating mainly underwent oxidative wear caused by its susceptibility to oxidation. To improve the stability of the MoN$_x$ coating at 500 °C, its oxidation resistance should be increased through elemental doping or other processes.

Figure 7 displays SEM images of the internal and external surfaces of the wear track of MoN$_x$_1# after rubbing at 400 °C. At 400 °C, large amounts of needle-rod and granular oxides were observed inside the wear track, whereas these oxides were present in smaller amounts outside the wear track (b), indicating that the formation of MoO$_3$ was promoted at 400 °C by rolling against the grinding ball. MoO$_3$ in the highly condensed state tended to behave as needle-like crystals [43]. At 500 °C, the oxide layer inside the abrasion marks (Figure 7c) was loose and porous, with large oxide particles. Meanwhile, a large region of glass phase formation was observed at the edge of the abrasion marks, likely formed because of the liquefaction of the oxide formed on the surface of the abrasion marks during the friction process and pushing to the edge of the abrasion marks due to squeezing against the abrasion edge. The oxide exhibited a flaky structure beyond the wear marks (Figure 7d), which was consistent with the morphology of $\alpha-$MoO$_3$, a loose structure that does not obstruct oxygen diffusion into the coating.

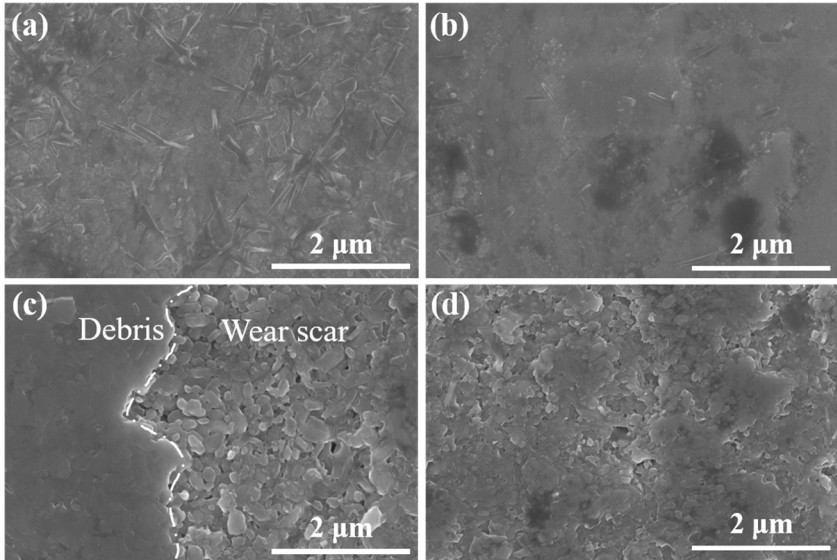

**Figure 7.** SEM surface images of (**a**,**c**) inside and (**b**,**d**) outside wear tracks of the MoN$_x$_1# coating after rubbing at (**a**,**b**) 400 °C and (**c**,**d**) 500 °C.

Figure 8 displays the Raman spectra for the interiors and exteriors of the wear scars of the (a) MoN$_x$_2#, (b) _3#, and (c) _4# coatings after the friction tests at 400 °C. In the Raman spectra, the MoO$_3$ peaks formed by the friction process were primarily located at 666, 820, and 996 cm$^{-1}$ [44]. The researchers induced the transformation of m$-$MoO$_2$ to $\alpha-$MoO$_3$ through continuous laser irradiation and observed Raman peaks corresponding to the crystalline phase of $\alpha-$MoO$_3$ at 152, 280, 660, 819, and 995 cm$^{-1}$. $\alpha-$MoO$_3$ exhibited a layered structure [40] that was susceptible to shear and thus promoted lubrication owing to the formation of layers that were bound to each other by van der Waals forces. Overall, the intensity of the MoO$_3$ peaks within the abrasion marks of the three groups of samples (4# > 3# > 1#) corresponded to the magnitude of the friction coefficient. In addition, more MoO$_3$ and oxygen-deficient oxides formed within the wear tracks of the MoN$_x$_4# sample, thereby resulting in its lowest COF. $\delta-$MoN was susceptible to oxidation at 400 °C [40], and the mixture of the high proportion of cubic phase with the low proportion of hexagonal phase underwent enhanced oxidation at 400 °C [41].

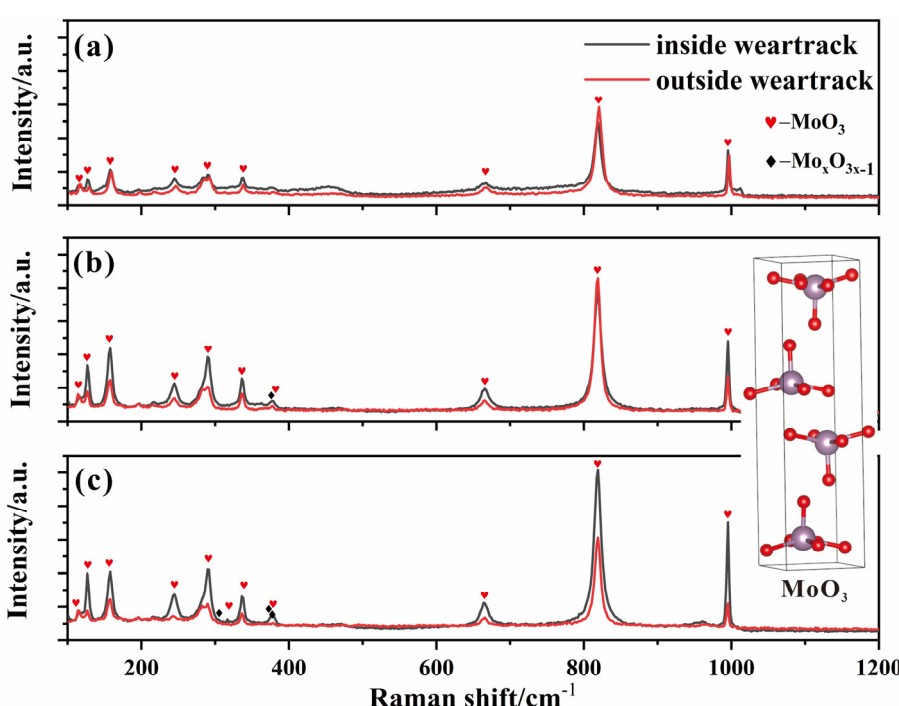

**Figure 8.** Raman spectra inside and outside the wear scars of the (**a**) $MoN_x$_2#, (**b**) _3#, and (**c**) _4# coatings after friction tests at 400 °C.

Figure 9a displays the GIXRD patterns of the $MoN_x$ coatings after rubbing at 400 °C. The $MoO_3$ peak was primarily located at 25.7°. No Magnéli (oxide) phase was formed outside the wear marks. The lowest intensity of the $MoO_3$ peak was observed in the spectra of $MoN_x$_1#, corresponding to its lower degree of oxidation. An intermediate oxide, $Mo_4O_{11}$, was formed only when the temperature was higher than 537 °C [45]. Therefore, after the friction test at 400 °C, $Mo_4O_{11}$ and other Magnéli phases were essentially absent from the GIXRD patterns for regions outside the abrasion marks and could be observed only inside the abrasion marks.

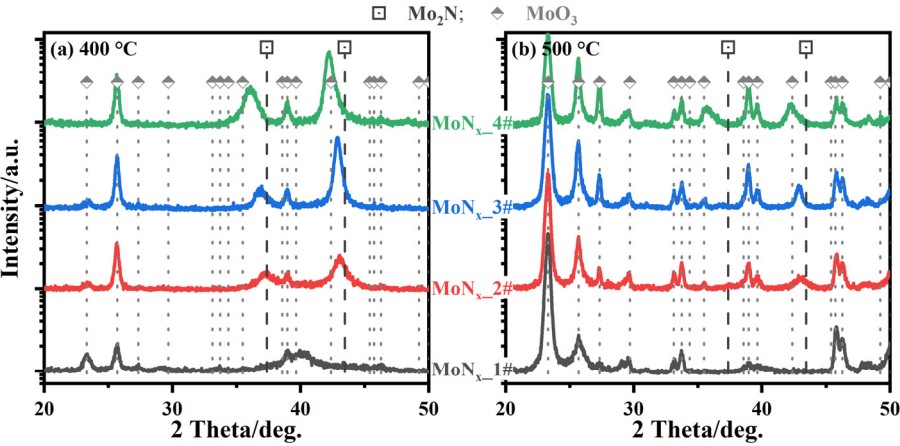

**Figure 9.** GIXRD patterns of $MoN_x$ coatings after rubbing at (**a**) 400 and (**b**) 500 °C.

Figure 9b displays the GIXRD patterns of the $MoN_x$ coatings after rubbing at 500 °C. The peak of nitride in $MoN_x$_1# basically disappeared, and the other samples still had weak $MoN_x$ peaks. The oxidation of $MoN_x$_1# and 2# at 500 °C was notably higher than that at 400 °C, and thus the COFs of both coatings exhibited decreases compared with those at 400 °C. The main diffraction peak of $MoO_3$ was observed at 23.3°, corresponding to $\alpha-MoO_3$. The differences in the COFs of the different $MoN_x$ coatings became smaller

as the friction proceeded (Figure 4c). The wear resistance of the $MoN_x$ coatings was poor because of the formation of a thicker oxide layer at 500 °C.

The tribolayer on the $MoN_x\_4\#$ coating after the ball-on-disc test at 400 °C was investigated through TEM and STEM, as displayed in Figure 10. A tribolayer with a thickness of approximately 100 nm on the $MoN_x$ coating was observed in the bright-field image (Figure 10a). Figure 10b displays the EDS compositional mapping and the EDS line-scan compositional profile of the tribolayer, indicating that the tribolayer mainly consisted of Mo and O. Figure 10c displays nanocrystalline grains A, which were essentially $MoO_3$ nanocrystals with (100) planes as the selective orientation and corresponded to $MoO_3$ with a needle-rod structure, as displayed in Figure 7a. Some poorly crystalline $MoO_3$ was observed around nanocrystals A. The $MoO_3$ did not tend to grow and dominated the entire friction layer, thereby forming a dense friction layer. This thin and dense friction layer blocked the diffusion of oxygen into the coating and thus caused the low wear of $MoN_x$ coatings at 400 °C. At 500 °C, a well-crystallized loosely structured porous friction layer was formed (Figure 7b), thereby resulting in reduced performance and severe oxidative wear of the $MoN_x$ coating, and the liquefaction of the oxide resulted in a low COF, but its adhesion led to an increased coefficient of friction.

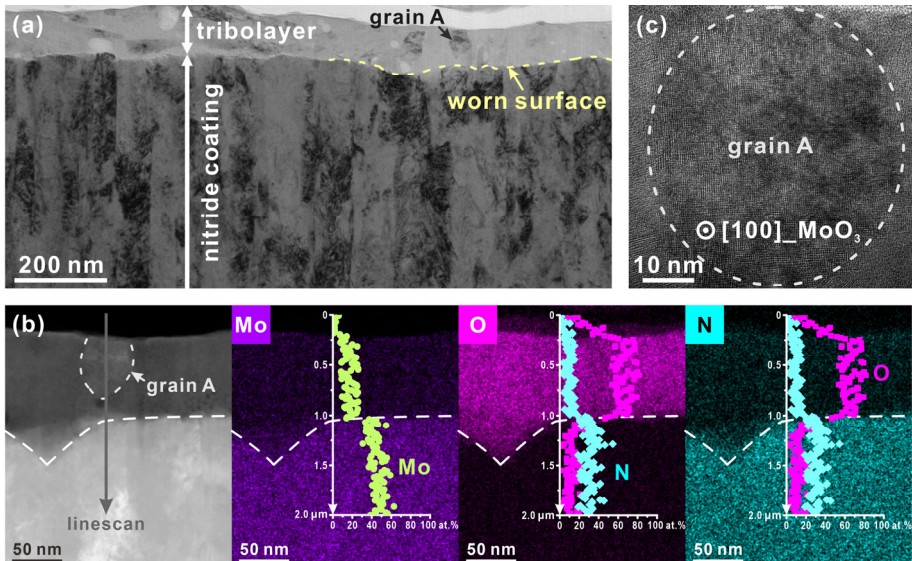

**Figure 10.** (**a**) TEM image of the cross-section inside the wear track for the $MoN_x\_4\#$ coating after the friction test at 400 °C. (**b**) STEM-HAADF image with EDS compositional mapping of the coating. (**c**) High-resolution TEM image of an oxide grain in the tribolayer on the $MoN_x\_4\#$ coating.

Figure 11a displays the hardness of the $MoN_x$ coatings after rubbing at different temperatures. It can be seen that the nanohardness of the coatings remained nearly constant after the friction test at 300 °C. The high hardness would help to maintain the good wear resistance of the $MoN_x$ coatings. The nanohardness of the sample surface notably decreased to approximately 10 GPa and 2 GPa after the friction tests at 400 and 500 °C, respectively. After the high−temperature friction tests, the hardness of $MoN_x\_2\#$ outside the abrasion marks was slightly higher than that of the other three samples. Figure 11b displays the Raman spectra for the interior of the wear marks of $MoN_x\_4\#$ rubbed at different temperatures. The Raman spectrum in the wear mark of the coating at 300 °C indicates that a little $MoO_2$ was formed in the wear mark. The $MoO_2$ had no lubricating effect, as shown by the $MoO_3$. This is another reason for the high COF of the $MoN_x$ coatings at 300 °C. It should be noted that $MoO_3$ appeared in the wear mark at 400 °C and 500 °C, and the content of $MoO_3$ increased notably with the increase in temperature from 400 to 500 °C, indicating that the $MoN_x$ coatings oxidized drastically. The coating ($MoN_x\_4\#$) could produce an appropriate amount of $MoO_3$ while maintaining a certain degree of

hardness at 400 °C and therefore showed low COF and wear rate values. Severe oxidation caused a drop in hardness around 500 °C, which also caused a rise in the coating wear rate.

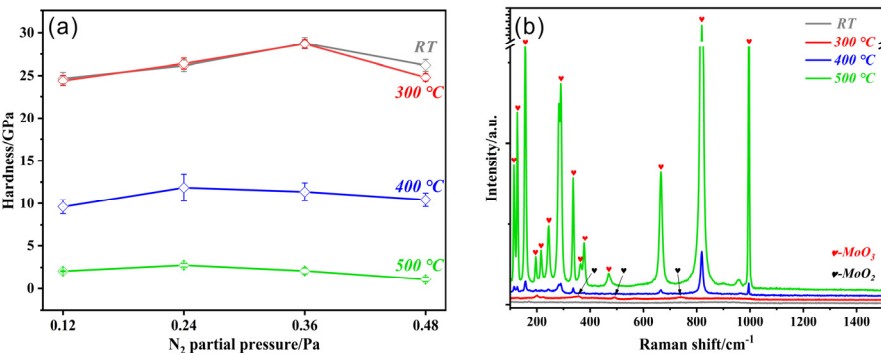

**Figure 11.** (**a**) Hardness of $MoN_x$ coatings after rubbing at different temperatures. (**b**) Raman spectra in the interior of the wear marks of $MoN_x$_4# rubbed at different temperatures.

In summary, the $MoN_x$ coatings formed $MoO_3$ and its compounds at RT with water during the friction process, thus the $MoN_x$ coatings exhibited advantages such as a low COF and a low wear rate at RT. At 300 °C, they had a high coefficient of friction due to the evaporation of water and the low oxidation of the coating itself, which was mostly $MoO_2$ formed inside the wear marks. The low wear rate was due to the filling of oxides in the wear marks. At 400 °C, the coating underwent low oxidation and formed Magnéli phases in the abrasion marks. The content of the $MoO_3$ and Magnéli phases had a large effect on the tribological properties. $MoN_x$_4# formed a dense friction film with a thickness of approximately 100 nm and lots of Magnéli phases in the abrasion marks; the friction film behaved as a wear-resistant friction-reducing layer and blocked the inward diffusion of oxygen so that the tribological performance of $MoN_x$_4# was optimal at 400 °C. At 500°C, the thicker oxide layer on the surface began to liquefy, which to some extent had a lubricating effect but also caused a significant rise in coating wear. The next step might be to lower the friction coefficient of $MoN_x$ coatings by including Cu and Ag components [23,46]. At both RT and high temperatures, the MoAgN [23] coating with magnetron sputtering in the literature had a lower friction coefficient than the $MoN_x$ coating in this paper, but its wear rate was noticeably higher, which also highlights the advancement of HiPIMS for the coating's wear resistance. The $MoN_x$ coatings' high wear rates at high temperatures did not improve, despite the doping of elements such as Ag, Cu, and V, which decreased the coating's friction coefficient. Instead, they became worse. As a result, it is more crucial to increase the wear resistance of $MoN_x$ coatings at high temperatures, and structural or coating modifications may be taken into account to increase the coatings' resistance to oxidation at high temperatures.

## 4. Conclusions

When prepared with HiPIMS under different $N_2$ pressures, the $MoN_x$ coating hardness was up to 28.8 GPa, and the coating phase structure was mainly fcc-$Mo_2N$. At low $N_2$ partial pressures, the coating phase structure was mostly $\alpha-Mo$, but at increased $N_2$ partial pressures, it was fcc$-Mo_2N$. Furthermore, the alleged $\delta-MoN$ was generated at a high $N_2$ partial pressure (0.48 Pa). At a $N_2$ partial pressure of 0.36 Pa, the coating reached its maximum hardness of 28.8 GPa. All $MoN_x$ coatings exhibited low COFs and wear rates at RT due to the formation of the compound $MoO_3/H_2O$, which behaved as a lubricant. However, the dry COFs of coatings at high temperatures (300 °C and 400 °C) were higher than those of coatings at RT. The increase in the dry COF was mainly due to the low level of oxidation and the evaporation of water. However, the $\delta-MoN$ phase formed in the $MoN_x$ coating deposited at the high $N_2$ partial pressure could facilitate the formation of $MoO_3$ and thus decrease the friction coefficient at 400 °C. As the friction test temperature further increased (500 °C), the growth rate of $MoO_3$ was very high, and the liquefaction of the

thicker oxide layer on the coatings exhibited desirable lubrication properties. However, serious oxidation caused a sharp increase in the wear rate. By optimizing the composition and structure of the coating, it is possible to reduce the dry friction of $MoN_x$ coatings by regulating the development rate of $MoO_3$ at varied high temperatures. The lubricating action of the $MoN_x$_4# tribolayer at 400 °C also prevented the coating from further oxidizing, which may offer a solution to the low wear of the $MoN_x$ coating at high temperatures.

**Author Contributions:** F.L.: Visualization and writing—original draft preparation. W.D. and Q.W.: conceptualization and methodology. W.D. and H.L.: Writing—reviewing and editing. Z.W.: Data curation. Q.W.: Funding acquisition. All authors have read and agreed to the published version of the manuscript.

**Funding:** This research was funded by the National Key Research and Development Program of China (grant No. 2017YFE0125400), and the financial support of the National Natural Science Foundation of China (grant No. 51875109) is acknowledged.

**Data Availability Statement:** The data used to support the findings of this study are available from the corresponding author upon request.

**Conflicts of Interest:** The authors declare no conflict of interest.

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
