# Peer review of "Dry Friction Performances of MoNx Coatings Deposited by High–Power Pulsed Magnetron Sputtering"

_magnetochemistry, doi:10.3390/magnetochemistry9030060_

Round 1

Reviewer 1 Report

In the manuscript with the title "Dry friction performances of MoNx coatings deposited by high power pulsed magnetron sputtering", the authors aim to study the dry friction performances of the MoNx coatings at different heating temperatures. The methods and discussion were well explained but the importance of this study is not well expressed. However, this manuscript can be considered for acceptance after the following minor points:

Line 81-82: The sentence "The findings of this study may contribute to the development of MoNx wear-protection coatings for mechanical components." would be better if it is located as the main important point expressed in the Abstract and/or Conclusion.

Line 109: Please consider assigning an equation number [such as '(1)' and Equation (1) in the text] for the equation of residual stress. At line 127, rewriting the Archard equation as Equation (2) would be better for recognizing the equations used in the study.

Line 146: Please consider assigning (by an arrow for instance) the thickness of the MoNx coating.

Line 172: If it is possible, please put a crystal model of different phases of Mo2N in Figure 2 to ease the reader in identifying the crystal structure. At Line 343, please put a crystal model of MoO3 to differentiate it from that of Mo2N.

There were recent published studies that cover the topic of mechanical coating, such as: doi.org/10.1016/j.mseb.2022.116066 (nanocomposite), doi.org/10.1016/j.jmapro.2023.01.037, doi.org/10.3390/jcs6120382, doi.org/10.1016/j.jallcom.2022.166592  (multilayer). Could the author compare the coating performance in the recent results with those of the previous researchers? This comparison could cover the hardness, wear rate, etc. The referee suggests that citing those references can improve the quality of this manuscript.

Author Response

Kindly check the attachment

Reviewer 2 Report

Major Comments:

1.       The Abstract may be modified in an impactful way as per the journal's reputation and demands.

2.       The introduction may also be improved by adding a few recent and nano-ranged metallic-based articles for the cleaning water study.

3.       The EDX spectra can be added in the SEM study to ensure the elemental contain (at%, table 1).

It will also help to identify the purity of the prepared sample.

4.       In XRD, what are the grain sizes? Authors could focus on size study in the case of GIXRD for example, https://doi.org/10.1038/s41598-018-20478-y

5.       In the TEM study also the author may find the atomic percentage of MoNx which can be well correlated with the XRD characteristic peaks for particular MoNx phases.

6.       The luminescence energy level diagram may also represent understanding the emission spectra on the basis of obtained spectral peaks and bad gap values. Fig. 6

7.       The application or testing part is fine as per the reported research work. But, in the results author may add more references to support the story of major findings.

8.       As per the given comments, try to modify the abstract and conclusion in an effective mode.

 Overall, the manuscript needs a major revision and then the revised version may consider for publication. 

Author Response

Kindly check the attachment

Round 2

Reviewer 2 Report

The authors have solved the comments in a proper way.

The revised version is suitable for publishing in the journal.